# Disproportionate exposure to urban heat island intensity across major US cities

Angel Hsu [1,2,3], Glenn Sheriff[4 ✉], Tirthankar Chakraborty [3,5] & Diego Manya[5]

Urban heat stress poses a major risk to public health. Case studies of individual cities suggest that heat exposure, like other environmental stressors, may be unequally distributed across income groups. There is little evidence, however, as to whether such disparities are pervasive. We combine surface urban heat island (SUHI) data, a proxy for isolating the urban contribution to additional heat exposure in built environments, with census tract-level demographic data to answer these questions for summer days, when heat exposure is likely to be at a maximum. We find that the average person of color lives in a census tract with higher SUHI intensity than non-Hispanic whites in all but 6 of the 175 largest urbanized areas in the continental United States. A similar pattern emerges for people living in households below the poverty line relative to those at more than two times the poverty line.

[1] Yale-NUS College, Singapore, Singapore. [2] School of Public Policy, University of North Carolina at Chapel Hill, Chapel Hill, NC, USA. [3] Data-Driven EnviroLab, Singapore, Singapore. [4] School of Politics and Global Studies, Arizona State University, Tempe, AZ, USA. [5] School of the Environment, Yale University, New Haven, CT, USA. ✉email: gsheriff@asu.edu

**B**uilt environments are commonly hotter than their neighboring rural counterparts[1]. This phenomenon, commonly referred to as the urban heat island effect, contributes to a range of public health issues. Heat-related mortality in the USA, for example, causes more deaths (around 1500 per year) than other severe weather events[2–4]. Heat exposure is also associated with several non-fatal health outcomes, including heat strokes, dehydration, loss of labor productivity, and decreased learning[5–12]. Characteristics of the built environment (e.g., green space, urban form, city size, spectral reflectance) not only create temperature differentials between urban and surrounding rural areas[13–16] but also contribute to intracity temperature variation[17–20]. This variation has the potential to cause disparities in the distribution of the burden of adverse heat-related outcomes across sociodemographic groups.

Like other environmental stressors, such as air pollution[21], low-income or otherwise marginalized communities may experience disproportionately higher levels of heat intensity[22]. Small-scale case studies have found disparities in the distribution of urban heat island intensity within single cities[23] or differences in exposure among population groups within a few cities in different countries[24–26]. Although evidence suggests that extreme heat-related morbidity and mortality in cities disproportionately affect marginalized groups[27–30], there has been little research showing whether these groups have systematic disproportionately high exposure to the heat island effect.

Instead, research linking intracity differences in heat exposure to sociodemographic factors has typically been done in an ad hoc manner for a small number of individual cities[23,29–32]. Examining the relationship between the distribution of annual urban heat island exposure and income at the neighborhood level, ref. [25] find that the distribution tended to favor those with higher incomes in 18 out of 25 selected global cities. While illustrative, these results are difficult to generalize since the sociodemographic information comes from a variety of sources with distinct definitions and methods, and the sample of global cities was chosen in response to data constraints rather than random sampling. It also does not convey information about potential disparities for other US cities.

In 108 US cities, ref. [26] find that neighborhoods that were redlined in the 1930s have summer surface temperature profiles that are significantly higher than other coded residential areas ("redlining" refers to the historical practice of denying home loans or insurance based on an area's racial composition). In light of substantial demographic changes and urban growth patterns over the past 90 years, however, the extent to which this finding translates into current racial or income disparities remains unclear.

While these studies are suggestive, it is difficult to extrapolate their results to a widespread or national level for several reasons. Varying methodological approaches to quantifying urban heat island intensity may lead to different conclusions, or analyses may not be representative. One obstacle to a more uniform approach has been the lack of consistent multicity delineations of urban and rural areas that are also comparable with the administrative areas of aggregation for which socioeconomic data are collected. Case studies may also reflect selection bias. Prior beliefs regarding inequitable distributions of heat exposure may have motivated such scientific inquiry for particular locations, such that the chosen cities may not be representative of the nation as a whole.

Combining high-resolution satellite-based temperature data with sociodemographic data from the US Census, we find that the average person of color lives in a census tract with higher summer daytime surface urban heat island (SUHI) intensity than non-Hispanic whites in all but 6 of the 175 largest urbanized areas in the continental United States. A similar pattern emerges for people living in households below the poverty line relative to those at more than two times the poverty line. In nearly half the urbanized areas, the average person of color faces a higher summer daytime SUHI intensity than the average person living below poverty, despite the fact that, on average, only 10% of people of color live below the poverty line. This last finding suggests that widespread inequalities in heat exposure by race and ethnicity may not be well explained by differences in income alone. While we do not observe major differences in SUHI intensity for very young or elderly populations in most major cities, when compared to the total population, we find that the same racial and ethnic disparities in SUHI for specific populations of color compared to non-Hispanic whites are also consistent for these age demographics.

## Results

Conceptually, an environmental risk analysis typically includes three components: hazard—measures of the spatial distribution of a potential harm; exposure—the intersection of the spatial distribution of human populations with the hazard; and vulnerability—the propensity to suffer damage when exposed to the hazard (see, for example, refs. [33,34]). We calculate harm on the basis of the census tract level database of SUHI intensity for the USA we developed in ref. [35]. During summer months, relatively large SUHI intensity is associated with increased local warming and extreme heat events in urban areas[13,36,37]. For exposure, we use census tract level demographic information from the 2017 5-year American Community Survey (ACS).

A comprehensive vulnerability assessment would require detailed information, not only about sociodemographic variables but also about other elements such as household resources, social capital, community resources, comorbidities, etc. that could be obtained at an individual or community level through localized fieldwork[38,39]. Although such an assessment is beyond the scope of this study, we consider one salient aspect, age, to evaluate whether differences in exposure by sensitive age groups affect conclusions drawn regarding exposure for the general population. In both very young and older populations, the body's ability to thermoregulate is compromised, and many older individuals have comorbidities or predispositions that increase the likelihood of heat-related illness and death[40,41]. Between 2004 and 2018, 39% of heat-related deaths in the USA occurred in ages 65 years or older[42]. Our framework is thus consistent with several studies using heat exposure to represent climate-related hazards and age to represent vulnerability to analyze the risk of heat stress in urban areas in Brazil, China, Finland, the Philippines, and the USA[34,43–46].

These combined data allow us to evaluate the relationship between race, income, age, and mean summer daytime SUHI intensity for all major urbanized areas in the USA (see "Methods" for the US Census definition of an urbanized area). These 175 largest US cities cover ~65% of the total population (see Supplementary Fig. 1) and are also where most US heat-related deaths have occurred in the last 15 years[42]. We narrow our analysis to the summer months of June, July, and August when the SUHI intensity is most pronounced during the day and when mean temperatures are generally higher than other periods through the year[47] (see Supplementary Fig. 2).

Recognizing that health impacts of summer heat exposure are likely to be nonlinear[48–51], i.e., incremental increases in environmental heat load may lead to disproportionately higher risk[47], we also consider environmental inequality metrics that evaluate the importance of within-group inequalities with respect to SUHI spatial distribution and exposure for different sociodemographic groups. We discuss our findings in three parts: first, comparing mean SUHI intensity across racial and income groups; second,

**Table 1 Mean summer daytime surface urban heat island intensity (SUHI) by climate zone and sociodemographic group.**

| | Climate zone (number of urbanized areas) | | | | |
| --- | --- | --- | --- | --- | --- |
| | **Arid** **(19)** | **Snow** **(44)** | **Temperate** **(110)** | **Equatorial** **(2)** | **Total** **(175)** |
| (a) Population-weighted means: Total | 0.40 (1.75) | 2.23 (2.71) | 2.21 (2.78) | 2.76 (2.20) | 2.06 (2.72) |
| By race/ethnicity[a]: People of color | 0.65 (1.61) | 3.44 (2.57) | 2.93 (2.74) | 3.19 (2.15) | 2.77 (2.70) |
| Hispanic | 0.74 (1.55) | 3.65 (2.72) | 3.03 (2.65) | 3.02 (2.19) | 2.70 (2.64) |
| Non-Hispanic Black | 0.74 (1.59) | 3.71 (2.33) | 3.04 (2.76) | 3.74 (1.91) | 3.12 (2.67) |
| Non-Hispanic White | 0.11 (1.86) | 1.67 (2.58) | 1.54 (2.65) | 1.93 (2.06) | 1.47 (2.60) |
| Non-Hispanic Other | 0.22 (1.78) | 2.68 (2.60) | 2.60 (2.84) | 2.34 (2.13) | 2.41 (2.80) |
| By income: Below poverty | 0.74 (1.61) | 3.32 (2.67) | 2.92 (2.78) | 3.42 (2.02) | 2.77 (2.73) |
| 1–2 × poverty | 0.69 (1.62) | 2.87 (2.69) | 2.64 (2.72) | 3.32 (2.03) | 2.50 (2.67) |
| Above 2 × poverty | 0.22 (1.79) | 1.87 (2.63) | 1.95 (2.76) | 2.41 (2.21) | 1.80 (2.69) |
| (b) Difference in means: People of color − Non-Hispanic white | 0.54*** (0.059) | 1.77*** (0.100) | 1.39*** (0.206) | 1.26*** (0.020) | 1.30*** (0.171) |
| Below poverty − 2 × poverty | 0.52*** (0.070) | 1.45*** (0.142) | 0.96*** (0.094) | 1.01*** (0.001) | 0.97*** (0.071) |
| People of color − below poverty | −0.10** (0.039) | 0.13* (0.071) | 0.02 (0.066) | −0.23 (0.042) | −0.00 (0.063) |
| Non-Hispanic white − below poverty | −0.63*** (0.070) | −1.65*** (0.094) | −1.38*** (0.167) | −1.50*** (0.022) | −1.30*** (0.127) |

Source: Author calculations, based on data from US Census Bureau and ref. [24]. Panel (a): Population-weighted means of urbanized area SUHI intensity in °C. Standard deviation is given in parentheses. Panel (b): Difference in group means. Standard errors clustered by urban area are given in parentheses.
*$p < 0.10$, **$p < 0.05$, ***$p < 0.01$.
[a]Hispanic is defined as all who report "Hispanic, Latino, or Spanish origin" as their ethnicity, regardless of race. People of color includes all Hispanic and all who do not identify as white alone. Black and white include all who identify as these races alone but not Hispanic. Other includes all other non-Hispanic races alone and more than one race.

using an inequality index to measure intragroup variation in SUHI intensity; and third, considering vulnerability according to age and race/ethnicity.

**Mean SUHI intensity across sociodemographic groups**. Table 1 (a) describes differences in exposure to SUHI by population groups defined by race/ethnicity and income (see "Methods" for demographic group definitions). We group urbanized areas by Köppen–Geiger[52] climate zones: arid, snow, warm temperate (henceforth referred to as temperate), and equatorial. For total population, summer day SUHI intensity is lowest (0.40 ± 1.75 °C) in arid zones, potentially due to the presence of more vegetation in urban areas compared to their rural references, which moderates the urban–rural temperature differentials[15,35]. Most cities are in snow and temperate zones, with a mean SUHI intensity of about 2.2 °C.

These population averages mask differences across population groups. With respect to race/ethnicity, in each climate zone, Black residents have the highest average SUHI exposure, for an overall average (±standard deviation) of 3.12 ± 2.67 °C, with Hispanics experiencing the second highest level (2.70 ± 2.64 °C). Non-Hispanic whites have the lowest exposure in each climate zone, with an overall average of 1.47 ± 2.60 °C. A similar pattern emerges across income groups: people living below the poverty line have the highest exposure in each zone (national average 2.70 ± 2.64 °C), while people living at above twice the poverty line have the lowest (1.80 ± 2.69 °C).

Figure 1 illustrates these sociodemographic differences in exposure, comparing kernel density plots of the distribution of mean SUHI across the 175 cities for different population groups. The starkest differences appear between race, Fig. 1a, and income, Fig. 1b. In only a few cities ($n = 17$) are white populations exposed to a mean SUHI intensity greater than 2 °C, while the corresponding number of cities for people of color is 83. A similar number of cities ($n = 82$) expose below-poverty populations to more than 2 °C SUHI. Figure 1c shows that distributions for those below poverty and for people of color are practically identical. As shown in Fig. 1d, e, there are not large differences in the distributions for the very young (less than 5) or the elderly (greater than 65) and the rest of the general population. Slightly more cities expose populations under 5 to higher SUHI intensity, while populations over 65 are exposed to lower mean SUHI intensity. Restricting attention to the most vulnerable age groups in Fig. 1g does not alter the conclusion drawn from Fig. 1a; for both age groups people of color appear to have a worse SUHI distribution than non-Hispanic whites.

Table 1(b) tests hypotheses that mean exposure is equal across selected groups. We reject ($p < 0.01$) both the null hypothesis of equal means for people of color and non-Hispanic whites in each climate zone, and the null hypothesis of equal means for people below and above two times the poverty line. Perhaps unsurprisingly, the average exposure of non-Hispanic whites is also significantly lower than the average exposure of people below poverty. Interestingly however, outside of arid climates, the

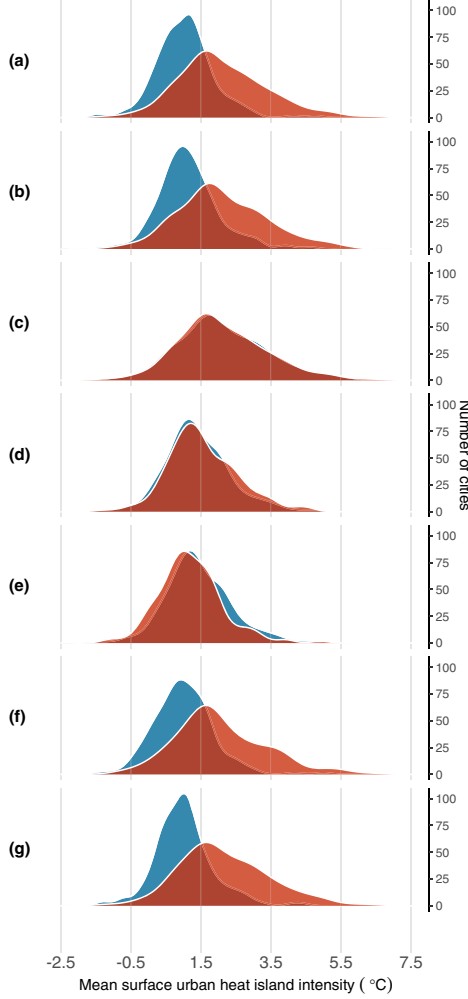

**Fig. 1 Distribution across cities of mean summer daytime surface urban heat island (SUHI) intensity by sociodemographic group.** Each panel compares kernel density estimates for two sociodemographic groups. Diagrams are normalized so that the area under each curve equals 175 cities. Hispanic is defined as all who report "Hispanic, Latino, or Spanish origin" as their ethnicity, regardless of race. People of color includes all Hispanic and all who do not identify as white alone. **a** Non-Hispanic white vs. all people of color. **b** 2× above poverty vs. below poverty. **c** Below poverty vs. all people of color. **d** Over 5 vs. under 5. **e** Under 65 vs. over 65. **f** Over 65: non-Hispanic white vs. all people of color. **g** Under 5: non-Hispanic white vs. all people of color. **a** illustrates that people of color have an average SUHI exposure greater than 2 °C in more cities than non-Hispanic whites.

average exposure of people of color is not significantly lower than the average exposure of people below poverty despite the fact that only 10% of people of color live below the poverty line.

The values in Table 1 are weighted by population, thus raising the possibility that a few exceptionally large urbanized areas may be driving the results. By illustrating the spatial distribution of significant city-level racial and income disparities in SUHI exposure, the maps in Fig. 2 visualize the geographic scope of the phenomenon presented in the table. For each comparison, circles and triangles identify which group has the higher average SUHI exposure in each city. Symbols with black outlines indicate cities for which the differences in means are statistically significant ($p < 0.05$). (Supplementary Table 1 displays city-level results used to generate these maps). In Fig. 2a, map shows that people of color have higher SUHI exposure than non-Hispanic

whites in 97% of cities nationally, and that this difference is significant in three quarters of cities. By zone, this proportion ranges from 42% in arid climates to almost 90% in snow. In contrast, non-Hispanic whites have a significantly higher exposure in only a single city, McAllen, TX. In Fig. 2b, the map shows a similar pattern for income. For over 70% of cities people below poverty have a significantly higher exposure than people above twice the poverty line (and in no city do they have a significantly lower exposure). In only 7% of cities nationwide does the average person of color have a lower exposure than the average person living below the poverty line (Fig. 2c).

**Intragroup variation in SUHI intensity**. A potential drawback to focusing on average exposures by demographic group is it can mask the existence of potential hotspots, geographic areas in which individuals are exposed to elevated levels of the hazard. Hotspots are particularly problematic when comparing exposures across groups if the additional damage caused by an incremental temperature increase grows as temperatures rise. In such cases, even if two groups were to hypothetically face the same average exposure, a group in which half of individuals were exposed to a temperature of, say, 38 °C and half were exposed to 32 °C, would suffer higher adverse effects than a group in which all individuals were exposed to 35 °C.

The Kolm–Pollak (KP) inequality index (see "Methods") is a tool for ranking group distributions of exposures when there are potential differences in dispersion of outcomes within each group (e.g., hotspots). Table 2(a) summarizes the average KP inequality index values for each city by population group and climate zone. A higher value corresponds to a less equal distribution of SUHI exposures within each group, with zero indicating a perfectly equal exposure (i.e., no within-group variation).

In general, cities in arid climates tend to have the lowest intragroup variation, and cities in snow and temperate zones have the highest. Within a given zone, however, index values are remarkably similar across population groups. Table 2(b) evaluates the hypothesis that index values vary significantly by demographic groups. Differences, measured in °C, are small in magnitude and not generally significant. Taken together, results in Table 2 suggest that the group means presented in Table 1 do not mask significant differences in variation within demographic groups. That is, the presence of relative hotspots is not likely to be higher among people living below the poverty line, for example, than people living at more than twice the poverty line. Consequently, for the remainder of this analysis we focus on average exposure levels for each group.

**Vulnerability**. Analyzing vulnerability is a relevant factor in considering the implications of the difference in mean exposures presented in Table 1. Since SUHI intensity is more damaging to people over the age of 65 years, the fact that all people of color might be exposed to higher average SUHI than non-Hispanic whites may not be problematic, for example, if its vulnerable (over 65) subpopulations are not exposed in the same way. Map in Fig. 2d indicates that people over 65 have lower SUHI exposures than those under 65 in 86% of US cities. While this difference is significant for only 16% of cities, there are no cities in which they have a significantly higher exposure. Table 3(a) presents mean SUHI exposure levels by race and ethnicity, restricting attention to two particularly vulnerable subpopulations: those over 65 years old and those below the age of 5 years. Comparing the exposure levels of these ages in Table 3(a) with group-wide exposure in Table 1(a), we see that for people of color exposure levels are nationally the same or higher for these vulnerable groups: 2.76 ± 2.64 °C for those below 5 and 2.88 ± 2.77 °C for

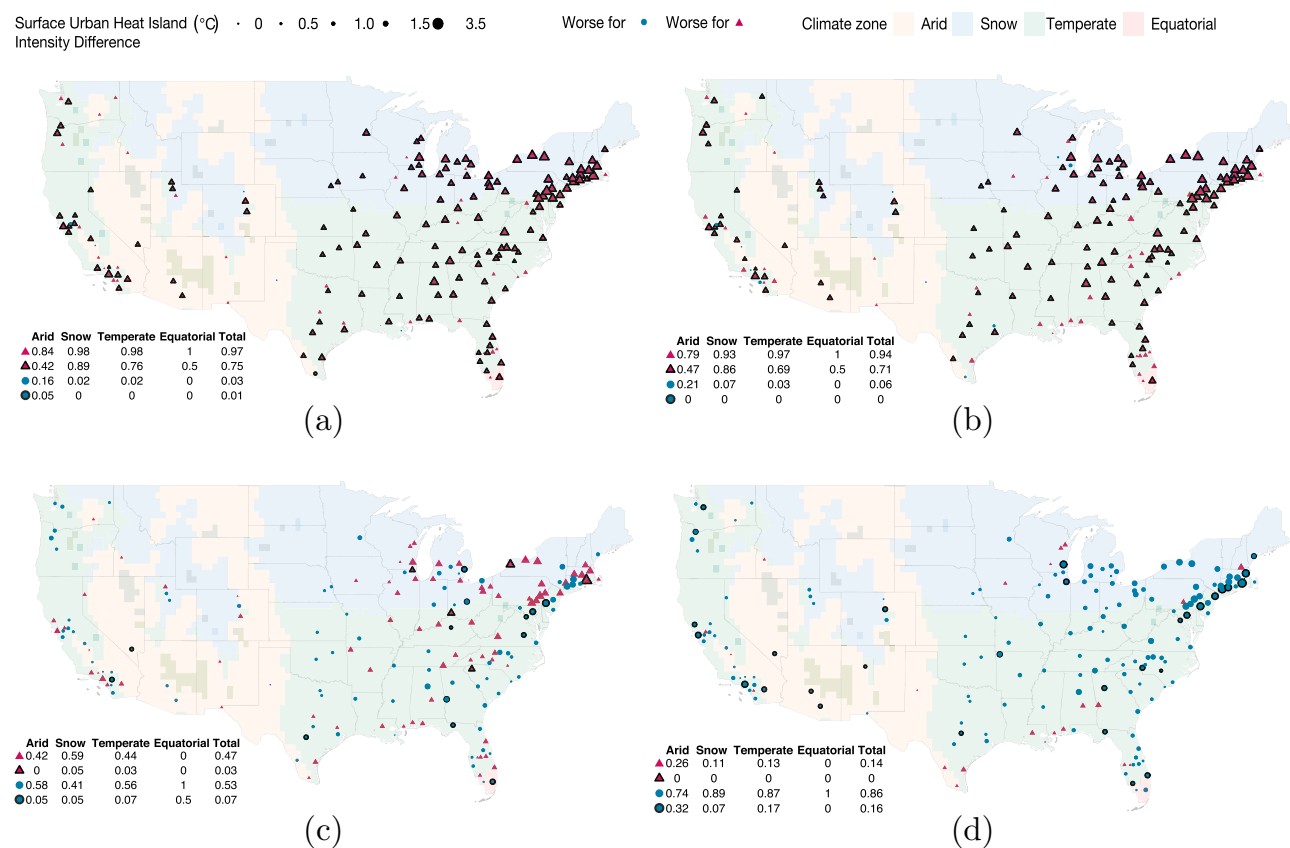

**Fig. 2 Sociodemographic differences in mean summer daytime surface urban heat island intensity by major urban area.** Symbols outlined in black depict statistically significant differences in mean exposures ($p < 0.05$). Tables embedded in the lower left-hand corners indicate proportion of cities in each category (e.g., worse for △ or worse for ○) by climate zone. Supplementary Table 1 provides detailed results for each city. Hispanic is defined as all who report "Hispanic, Latino, or Spanish origin" as their ethnicity, regardless of race. People of color includes all Hispanic and all who do not identify as white alone. **a** Non-Hispanic white (○) and people of color (△). **b** Above 2 × poverty (○) and below poverty (△). **c** Below poverty (○) and people of color (△). **d** Below 65 (○) and above 65 (△).

those above 65, compared to 2.77 ± 2.70 °C for all people of color. For non-Hispanic whites, however, these vulnerable populations have slightly lower exposures: 1.45 ± 2.53 °C for those below 5 and 1.44 ± 2.60 °C for those above 65, compared to 1.47 ± 2.60 °C for the entire white population. Table 3(b) compares mean exposures of these vulnerable ages across racial/ethnic groups. The patterns are almost identical to results in Table 1(b): people of color in each age group have significantly higher exposure levels than their white peers in each climate zone.

## Discussion

**Framework for understanding inequalities in SUHI.** This analysis provides a framework for quantifying the intercity and intracity distribution of SUHI intensity by race, income, and age that considers both the intensity of the exposure as well as the inequality of distribution for different population subgroups. We find that the distributions of summer daytime SUHI intensity, taking into account both the mean and dispersion, is worse for both people of color and the poor, compared to white and wealthier populations in nearly all major US cities. As illustrated in Fig. 2, this pattern holds not only at the national level, but in almost all major urban areas regardless of geographical location or climate zones, with a particularly intense difference in the Northeast and upper Midwest of the continental United States. These findings provide comprehensive evidence supporting the narrative presented by earlier case studies that minority and low-

income communities bear the brunt of the urban heat island effect[23,25,26,29–32,35], air temperature[23], and heat stress[31] in individual or multicity studies.

Although age presents a vulnerability to SUHI, and elderly individuals aged 65 and older comprise a substantial percentage (39%) of heat-related deaths in the USA[42], our finding that populations over 65 are on average slightly less exposed (1.84 °C versus 2.06 °C for those under 65) could have several explanations. Because SUHI intensity and greenness (as measured by normalized difference vegetation index) are negatively correlated[35], cooler areas tend to be greener. There is evidence that populations over the age of 65 tend to live in suburban areas in the USA. Approximately half live in rural areas or in urban areas with less than 1 housing unit per acre, and 28% live in suburban areas[53], which are typically greener than denser urban areas, except in arid climates[15,54,55]. Considering the intersection of race and age demographics, however, the same racial and ethnic disparities in SUHI intensity for specific populations of color compared to non-Hispanic whites are also consistent for both very young and elder populations[3], meaning non-white populations over the age of 65 or less than 5 are still exposed to higher levels of SUHI than their white counterparts. The fact that older people of color have a slightly higher SUHI exposure than all people of color suggests that they may be less able to escape the heat by changing location than their white counterparts.

The Intergovernmental Panel on Climate Change has identified the "increasing frequency and intensity of extreme heat, including

**Table 2 Kolm–Pollak inequality index of summer daytime surface urban heat island intensity (SUHI) by climate zone and sociodemographic group.**

| | Climate zone | | | | |
| --- | --- | --- | --- | --- | --- |
| | **Arid** | **Snow** | **Temperate** | **Equatorial** | **Total** |
| (a) Population-weighted index means: Total | 0.12 | 0.29 | 0.27 | 0.20 | 0.26 |
| | (0.09) | (0.11) | (0.12) | (0.03) | (0.13) |
| By race/ethnicity[a]: People of color | 0.10 | 0.24 | 0.23 | 0.19 | 0.22 |
| | (0.07) | (0.08) | (0.12) | (0.02) | (0.11) |
| Hispanic | 0.09 | 0.25 | 0.21 | 0.20 | 0.19 |
| | (0.06) | (0.08) | (0.11) | (0.02) | (0.11) |
| Non-Hispanic Black | 0.09 | 0.19 | 0.22 | 0.15 | 0.21 |
| | (0.05) | (0.07) | (0.08) | (0.01) | (0.08) |
| Non-Hispanic White | 0.14 | 0.27 | 0.27 | 0.18 | 0.26 |
| | (0.12) | (0.11) | (0.12) | (0.04) | (0.12) |
| Non-Hispanic Other | 0.13 | 0.25 | 0.27 | 0.20 | 0.26 |
| | (0.08) | (0.11) | (0.17) | (0.03) | (0.16) |
| By income: Below poverty | 0.10 | 0.25 | 0.24 | 0.17 | 0.23 |
| | (0.08) | (0.10) | (0.11) | (0.02) | (0.11) |
| 1–2 × poverty | 0.10 | 0.26 | 0.24 | 0.17 | 0.22 |
| | (0.08) | (0.11) | (0.11) | (0.02) | (0.11) |
| Above 2 × poverty | 0.13 | 0.28 | 0.27 | 0.21 | 0.26 |
| | (0.10) | (0.11) | (0.13) | (0.04) | (0.13) |
| (b) Difference in mean index values: People of color — Non-Hispanic white | −0.04 | −0.04 | −0.04 | 0.01 | −0.04* |
| | (0.055) | (0.031) | (0.030) | (0.018) | (0.023) |
| Below poverty — 2 × poverty | −0.03 | −0.02 | −0.03 | −0.04 | −0.03 |
| | (0.048) | (0.032) | (0.029) | (0.014) | (0.023) |
| People of color — below poverty | 0.00 | −0.02 | −0.01 | 0.02* | −0.01 |
| | (0.038) | (0.027) | (0.026) | (0.007) | (0.020) |

Source: Author calculations, based on data from US Census Bureau and[24]. Panel (a): Population-weighted mean of urban area Kolm–Pollak indexes in °C with moderate inequality aversion. Standard deviation is given in parentheses. Panel (b): Difference in group means. Robust standard errors are given in parentheses.
*$p < 0.10$.
[a]Hispanic is defined as all who report "Hispanic, Latino, or Spanish origin" as their ethnicity, regardless of race. People of color includes all Hispanic and all who do not identify as white alone. Black and white include all non-Hispanics identifying as these races alone. Other includes all other non-Hispanic races alone and more than one race.

the urban heat island effect" as a relevant hazard for certain age groups (i.e., elderly, the very young, people with chronic health problems), which creates a risk of increased morbidity or mortality during extreme heat periods[37]. Relating intercity SUHI disparities to health outcomes is challenging due to both prevalence of confounding factors in the populations groups, as well as the differences between land surface temperature (LST) and more comprehensive metrics of heat stress[56]. There is, however, evidence of disparities in heat-related health outcomes across the USA and for individual cities[42,57]. For example, ref. [57] finds positive correlations between heat-related mortality rates and poverty for neighborhoods in New York City. More recently, ref. [42] found higher heat-related mortality rates among non-Hispanic American Indians/Alaska Natives and Blacks than for non-Hispanic whites at the national level.

**Locally-tailored SUHI mitigation strategies**. In addition to evaluating the general scope of potential heat-related environmental inequality concerns, the metrics developed in our study can identify precisely in which cities specific sociodemographic groups are most adversely exposed to SUHI intensity and to potential heat-related health effects for vulnerable groups. These data can thereby assist policy makers in designing interventions to address this exposure differential, as well as facilitate analysis of different scenarios to select the most appropriate strategy to mitigate exposure in an equitable manner. According to ref. [47], many cities do not take into consideration the spatial location of the most exposed populations in climate mitigation planning and whether areas that present increased sociodemographic vulnerabilities, such as age or high minority populations, are coincident with areas exposed to higher temperatures.

Consideration of background climate differences, which have been found to strongly modulate the thermodynamics of SUHI intensity[15,16], are critical for adapting city-specific intervention strategies to reduce both total exposure and disparities in its distribution[58]. Because we use a globally consistent dataset derived from satellite remote sensing[35], our data allow for comparison of SUHI given differences in background climates and sociodemographics. Decision-makers and urban planners can utilize this information as a starting point to identify best practices and strategies for mitigating the overall SUHI as well as inequalities in its distribution, although there are certainly localized, context-specific factors that must be considered when determining SUHI management strategies. Studies have demonstrated the importance of coproduction (i.e., involving citizens in the production of knowledge and planning decisions) in developing tailored urban environmental policies[59]. Manoli et al.[60], who used similar globally consistent satellite-derived data to evaluate drivers of SUHI in 30,000 cities around the world, acknowledge that these data can provide a first-order analysis to understand base-level SUHI exposures and differences to complement more fine-grained data on local factors that influence the SUHI (see "Study limitations" section for more discussion on data issues).

For example, the presence (or absence) of urban vegetation is often proposed as a strategy to reduce the urban heat island effect[13,16,20,61], climate change more generally[62], and for their other cobenefits[63–66]. Access to green space has been found to be inversely correlated with median income[67]. Actions such as planting trees in low-income and minority neighborhoods, which has been shown to reduce summertime afternoon temperatures by as much 1.5 °C[68], can increase property values and housing

**Table 3 Mean summer daytime surface urban heat island intensity (SUHI) by climate zone and age.**

| | Climate zone | | | | |
| --- | --- | --- | --- | --- | --- |
| | Arid | Snow | Temperate | Equatorial | Total |
| (a) Population-weighted means—Below 5 years old: Total | 0.55 | 2.38 | 2.37 | 2.94 | 2.20 |
| | (1.67) | (2.66) | (2.73) | (2.16) | (2.68) |
| People of color[a] | 0.73 | 3.41 | 2.94 | 3.24 | 2.76 |
| | (1.57) | (2.53) | (2.68) | (2.06) | (2.64) |
| Black[b] | 0.85 | 3.81 | 3.13 | 3.82 | 3.21 |
| | (1.54) | (2.26) | (2.72) | (1.82) | (2.62) |
| Hispanic[c] | 0.81 | 3.58 | 3.01 | 3.01 | 2.69 |
| | (1.53) | (2.66) | (2.62) | (2.11) | (2.60) |
| Non-Hispanic white[d] | 0.16 | 1.59 | 1.53 | 1.88 | 1.45 |
| | (1.80) | (2.49) | (2.59) | (2.16) | (2.53) |
| Above 65 years old: Total | 0.16 | 2.03 | 1.96 | 2.58 | 1.84 |
| | (1.82) | (2.66) | (2.79) | (2.19) | (2.72) |
| People of color[a] | 0.55 | 3.58 | 3.01 | 3.38 | 2.88 |
| | (1.62) | (2.54) | (2.82) | (2.13) | (2.77) |
| Black[b] | 0.69 | 3.82 | 3.22 | 3.77 | 3.28 |
| | (1.63) | (2.33) | (2.83) | (1.92) | (2.72) |
| Hispanic[c] | 0.65 | 3.85 | 3.16 | 3.32 | 2.80 |
| | (1.53) | (2.79) | (2.70) | (2.16) | (2.68) |
| Non-Hispanic white[d] | −0.02 | 1.69 | 1.51 | 1.91 | 1.44 |
| | (1.87) | (2.57) | (2.66) | (2.01) | (2.60) |
| (b) Difference in means—Below 5 years old: People of color[a] − Non-Hispanic white[d] | 0.57*** | 1.82*** | 1.41*** | 1.36*** | 1.31*** |
| | (0.078) | (0.106) | (0.159) | (0.018) | (0.138) |
| Above 65 years old: People of color[a] − Non-Hispanic white[d] | 0.57*** | 1.88*** | 1.50*** | 1.47** | 1.44*** |
| | (0.086) | (0.111) | (0.258) | (0.080) | (0.209) |

Source: Author calculations, based on data from US Census Bureau and ref. [35]. Sample includes all urbanized areas with 2017 population over 250,000. Panel (a): Population-weighted means of urbanized area SUHI intensity in °C. Standard deviation is given in parentheses. Panel (b): Difference in group means. Standard errors clustered by urban area are given in parentheses. alone.
**p < 0.05, ***p < 0.01.
[a]People of color includes all Hispanic and all who do not identify as white alone.
[b]Black alone, including Hispanic black.
[c]Defined as all who report "Hispanic, Latino, or Spanish origin" as their ethnicity, regardless of race.
[d]Non-Hispanic white alone.

costs. Previous work indicates that these housing price effects may displace minority residents the policies were designed to help[69,70]. Evidence suggests that homeowners value cooler temperatures and that local temperature differentials are capitalized into housing prices[71]. It is therefore unsurprising that people living below the poverty line have higher average temperature exposures than those at over two times above the poverty line in 94% of major urbanized areas in our study.

**Complexity in disentangling race, income, and SUHI.** The effect of historical practices of real estate, urban development, and planning policies that promoted spatial and racial segregation in US cities[26,72], as well as the fact that people of color tend to have lower income than white populations in the USA makes it difficult to disentangle purely economic reasons for the unequal distribution of SUHI intensity exposure to those based upon racial factors. We can, however, shed light on the complex relationships between race, poverty, and urban heat by comparing the SUHI distributions faced by people of color to those faced by people living below the poverty line.

While there is some overlap of individuals belonging to both groups, such individuals are a minority; according to the 2017 5-year ACS, only about 10% (ranging from 0.4 to 18.9%) of people of color live below the poverty line in these major urbanized areas. If income were to determine local summer daytime SUHI intensity exposure, one would expect that the typical person of color would have a lower exposure than the typical person living below poverty. Table 1 shows that this hypothesis is unsupported: across the entire sample the mean SUHI exposure of a person of color (2.77 ± 2.70 °C) is practically identical to that of a person

living below poverty (2.77 ± 2.73 °C). The distribution of temperature differentials across cities is also similar for these two groups (Fig. 1). Nationally, we observe few cities (about 10%) with statistically significant differences between the mean SUHI intensities for these groups (Fig. 2c).

**Illustrative examples.** While the SUHI distributions for below poverty and people of color are nearly identical (Fig. 1), patterns of exposure by sociodemographic group are not all the same between cities. Figure 3 provides an illustrative example, contrasting the cases of Baltimore, MD, and Greenville, SC. In Baltimore, the temperature exposure of the average person of color is about 0.7° cooler than the average person in poverty, whereas the opposite is true for Greenville. Figure 3a, b shows that in Greenville, the Black population is highly concentrated in the warmest census tracts, while the poor population is more widely dispersed to cooler areas away from the city center. In Baltimore by contrast, Fig. 3c, d indicates that the poorest census tracts tend to be the warmest, while the Black population is much more evenly spread through the city.

As these illustrative examples of Greenville, SC, and Baltimore, MD, show, while many factors might explain our observed difference in below poverty and minority populations' SUHI exposure in these two cities, prior research on residential housing markets in the USA has shown that racial and ethnic segregation, among factors other than consumer preference alone, determine where certain groups live[73,74].

**Future challenges.** The patterns of systematically higher SUHI exposure for low-income populations and communities of color

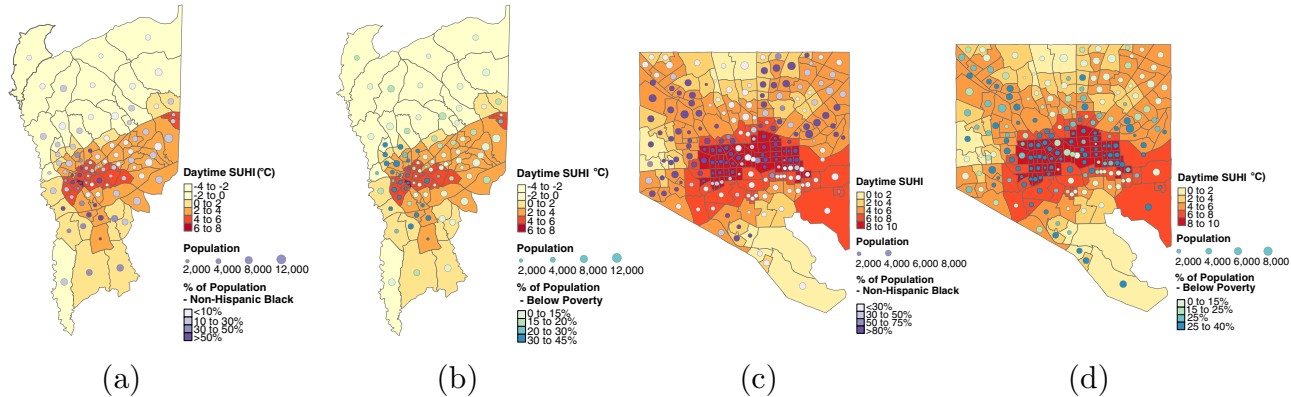

**Fig. 3 Distribution of surface urban heat island intensity (SUHI) by race and income in Greenville, SC, and Baltimore, MD.** The correlation between SUHI intensity (dark orange and red) and census tracts that are predominantly non-Hispanic Black (in dark purple) and low-income areas (in dark teal) differs across cities. Hispanic is defined as all who report "Hispanic, Latino, or Spanish origin" as their ethnicity, regardless of race. **a** Greenville, SC: SUHI and race. **b** Greenville, SC: SUHI and income. **c** Baltimore, MD: SUHI and race. **d** Baltimore, MD: SUHI and income.

in nearly all major US cities may lead to further inequality if these disparities persist or worsen. Currently disadvantaged groups suffer more from greater heat exposure that can further exacerbate existing inequities in health outcomes and associated economic burdens, leaving them with fewer resources to adapt to increasing temperature[75]. Increasing trends of urbanization, demographic shifts with aging populations, and the projected rise in extreme heat-related events due to climate change[37], may compound certain groups' vulnerability to extreme heat in the future[29,38]. From an environmental equity and justice perspective, understanding where these disparities in heat exposure exist today can inform future efforts to design policy interventions to ameliorate them.

**Study limitations.** While the SUHI database used in this study has been validated against other published estimates[35], we recognize limitations of its use as a metric to identify which groups may be more vulnerable to heat stress within cities. Our environmental equity analysis assumes that SUHI intensity is harmful. While this assumption is likely to be justified in the summer periods evaluated in this study, the effect may be beneficial in cities exposed to extreme winter cold[76]. Although in theory the association between SUHI intensity and income and race could imply less extreme cold-related stress in poorer and predominantly non-white neighborhoods, other research suggests that these winter benefits may not materialize[35]. Nonetheless, intracity variation should be taken into account while planning strategies both to reduce mean SUHI and to address environmental disparities in its exposure within cities.

Heat stress also depends on factors other than LST and air temperature, including humidity, wind speed, and radiation[77]. SUHI intensity, however, is still a useful proxy for the urban contribution to local heat stress[35]. Our analysis relies on satellite-based estimates, which could overestimate UHI magnitude compared to in situ weather stations, particularly during daytime[78], when shade from tree canopies or buildings reduce air temperature in a way that is not captured from a satellite's vantage point. Our estimates, therefore, likely slightly overestimate the absolute measures of UHI (in °C), but in lieu of dense, widely accessible ground-based air temperature networks, satellite-derived estimates represent the best available data source.

We assume every individual residing in a census tract has the same temperature exposure. In reality, temperatures and demographic characteristics may vary within a tract, and exposures can

depend on individual behavior or conditions (home air conditioning, time spent outdoors, etc.). Our analysis also assumes that people pass the entire day in their census tract, abstracting from the possibility that they spend work or leisure time in other locations with distinct SUHI profiles.

The choice to use census tract as the unit of analysis is a compromise based on the relative precision of demographic and satellite data. Precise demographic data are publicly available at the smaller census block group level, and aggregating to larger tracts implies a loss of information. In other contexts, the environmental justice literature suggests that such aggregation can underestimate racial disparities due to the "ecological fallacy"[79]. In contrast, although satellite data are available at a resolution of 1 km, this pixel-level data have a relatively high degree of uncertainty, particularly for urban areas[80]. Since census tracts, unlike block groups, typically contain more than one pixel, averaging the satellite data to this level of aggregation provides more reliable surface temperature estimates.

We also do not evaluate inequities in SUHI among demographic groups over time. Future research could evaluate whether disparities in SUHI exposure have improved or worsened in time. A recent study examining inequality in fine particulate air pollution ($PM_{2.5}$) found that between 1981 and 2016, absolute disparities between more and less polluted census tracts in the USA declined but that relative disparities have persisted, meaning the most exposed subpopulations in 1981 remained the most exposed in 2016[81]. Incorporating a time-series panel dataset on SUHI intensity and sociodemographic characteristics would allow for future understanding of the role climate change and increasing temperatures may have on worsening heat exposure disparities over time.

## Methods
**SUHI intensity database.** Existing maps of SUHI intensity use physical boundaries (e.g., boundary based on built-up, impervious land cover usually measured through satellite remote sensing) as the units of calculations for delineating both urban areas and their corresponding rural references, making them unsuitable for use with socioeconomic data without significant uncertainties. To deal with this scale mismatch between administrative and physical boundaries, we use summertime (June, July, and August; Supplementary Fig. 1) values from our recently created SUHI database for the USA that is consistent with census tract delineations[35].

This dataset uses global LST products from NASA's MODIS sensor[82] and the land cover product from the European Space Agency[83]. It calculates SUHI intensity at the census tract level by combining the land cover data with the census tracts that intersect US urbanized areas, as defined by the US Census Bureau[84].

We use the simplified urban extent method[15] to define the SUHI intensity of an urban census tract $t$ as the difference between the tract's mean LST and the mean

temperature of the rural reference $r$, the nonurban, nonwater land cover pixels within the tract's urbanized area

$$\mathrm{SUHI}_t = \mathrm{LST}_t - \mathrm{LST}_r. \qquad (1)$$

Urbanized area boundaries do not necessarily coincide with those of census tracts. In such cases, we adjust the approach to include only pixels within the urbanized area of a census tract to calculate $\mathrm{LST}_t$. For more details, see ref. [35]. The distributional analysis thus implicitly assumes no one resides in the nonurbanized portions of those outlying tracts.

Since previous studies have demonstrated the importance of background climate in modulating the SUHI intensity[15,16], we also examine the relationship between disparities in SUHI exposure and the Köppen–Geiger climate zone[85]. The possible impact of background climate has policy implications, since it constrains what city planners can do to mitigate the city-specific SUHI and its distributional impacts.

**Demographic data**. We assign the same SUHI intensity to every individual living in a given census tract. Demographic group averages are calculated as weighted means across census tracts, in which the weights correspond to the number of people of a given group residing in a tract. Census tract level demographic data come from the 2017 ACS 5-year Data Profile[86,87]. We collect data on race, ethnicity, poverty status, age, and age by race for all 46,346 census tracts in the 175 census-defined urbanized areas that contain more than 250,000 residents (Supplementary Fig. 2). Our set of urbanized areas ranges from 43 to 4470 tracts, with a median of 582 (Supplementary Table 2). Responses to race include options for single race (e.g., Black only) as well as multiple races. Hispanic is an ethnicity reported in addition to race (e.g., Black only and Hispanic). Regardless of race, it is defined as any who respond "yes" to the Census question asking whether the person is "of Hispanic, Latino, or Spanish origin"[88]. For the total population, we generate categories for two non-Hispanic single race groups (Black, white), Hispanic of any race, and "Other". Other includes non-Hispanics of other single races, including Black or African American, Asian, American Indian and Alaska Native, Native Hawaiian and other Pacific Islander, and non-Hispanics reporting two or more races. We also create a People of Color category that includes all Hispanic and all who do not identify as white alone. For age categories, we use the same race and ethnicity groupings to develop under 5 and over age 65 categories. Since ACS age data do not differentiate Black by Hispanic ethnicity, however, Black Hispanics appear in both the Black and Hispanic categories in Table 3 only.

The ACS reports poverty status as household income relative to the poverty line. This income is not measured in dollars since the poverty line depends on the number of individuals in the household. We use these data to generate three income categories: at or below the poverty line, from one to two times the poverty line, and at or above two times the poverty line (the highest recorded category). While results for each of these income categories are provided in our tables, for the ease of exposition, we focus our discussion on the tails of the income distribution: the poor (those below poverty) and the relatively rich (above two times).

**Inequality metrics**. The goal of comparing exposure levels across population groups is to determine whether a distribution of SUHI intensities for a given group is preferable in some sense to that of another. In contrast to approaches identifying correlations between summer temperatures and neighborhood characteristics such as historical redlining[26] or percentage poor or low income, e.g., ref. [23], we place the unit of analysis on the individual to better understand human welfare implications of SUHI exposure.

There is no clear link between what individuals find desirable and the significance of statistical correlations between neighborhood attributes. It is theoretically possible, for example, for the average individual in a demographic group to be better off with a positive (versus negative) correlation between summer heat and their group's majority status in a neighborhood if most members of the group happen to live in neighborhoods in which they are a minority.

A simple individual-based metric such as mean exposure is potentially misleading due to nonlinear adverse health impacts of summer heat. Evidence suggests that above a moderate threshold damage is an increasing convex function of temperature, i.e., a 1° temperature increase causes more damage at higher temperatures[48–51]. In such cases, Jensen's inequality implies that, all else equal, the average health damage for a population in which everyone faces an identical summer heat exposure will be lower than that of a population with the same mean exposure but an unequal temperature distribution. It follows that for any unequal temperature distribution there exists a more desirable (from a health perspective) distribution characterized by a higher mean and no inequality. That is, a perfectly equal summer temperature distribution is generally preferable to an unequal distribution with the same mean.

Using this principle, we adapt an ethical framework commonly used to study income distributions to compare distributions of environmental harm[89]. Under this framework, a distribution is considered more desirable than another if it would be chosen by an impartial agent who knows only that she will receive an outcome from that distribution but is ignorant regarding what that outcome will be. Reframing the problem of ranking SUHI exposure distributions as one of rational choice made behind a "veil of ignorance"[90,91], provides an intuitive approach founded on explicitly specified individual preferences.

To implement this method, we transform distributions of SUHI intensity across individuals in a demographic group to "lotteries" in which the probability of receiving a given exposure corresponds to the proportion of people in the group receiving that exposure. The more desirable distribution is the lottery that would be chosen ex ante by an impartial representative agent who only knows that her ex post exposure will be randomly drawn from that lottery. This choice in turn depends on assumptions made about the agent's tastes regarding the harm caused by different levels of exposure.

The equally distributed equivalent (EDE)[92,93] is a construct for cardinally ranking all possible lotteries. It represents the value of the outcome (in our case, SUHI intensity) that, if experienced by everyone in the group, would make the impartial agent indifferent between the actual unequal distribution and the hypothetical equal distribution.

In summer, the EDE is generally higher than the mean of the actual distribution, i.e., the agent would be willing to bear a higher average intensity if she knew that she were guaranteed not to randomly draw a value higher than the mean[89]. The gap between the EDE and the mean is an index of inequality within a given group, indicating the maximum additional SUHI intensity per person that would make the representative agent indifferent between the actual distribution and the hypothetical equal distribution.

As described in ref. [89] and Supplementary Note 1, the KP inequality index has several desirable features relevant to characterizing distributions of environmental harm. For an $N$-dimensional vector of SUHI intensities $\mathbf{x}$, with each element corresponding to the exposure of individual $n$ in a given urbanized area, the KP inequality index can be expressed

$$I(\mathbf{x}) = -\frac{1}{\kappa}\ln\frac{1}{N}\sum_{n=1}^{N}e^{\kappa[\bar{x}-x_n]}, \text{ for } \kappa < 0. \qquad (2)$$

Here, $\bar{x}$ is the mean outcome and $\kappa$ is a parameter indicating the degree to which inequality in the distribution is undesirable due to increasing marginal damage. The KP EDE is simply $I(\mathbf{x}) + \bar{x}$. As is standard in the literature, we present results for a range of possible values for $\kappa$ (see Supplementary Tables 3–5).

**Software**. All statistical analyses were conducted in Stata (Version 15) and R (Version 3.6.3). Figures were made using ggplot2[94] and tmap[95,96] packages in R. The SUHI dataset was created using the Google Earth Engine platform[97].

**Reporting summary**. Further information on research design is available in the Nature Research Reporting Summary linked to this article.

## Data availability
SUHI intensity data are available for exploration on an interactive Google Earth Engine platform tool, available at https://datadrivenlab.users.earthengine.app/view/usuhiapp and also for download at https://data.mendeley.com/datasets/x9mv4krnm2/2. Sociodemographic data were collected from the US Census Bureau 2017 5-year ACS via the API at https://api.census.gov/data/2017/acs/acs5/variables.html.

## Code availability
Code to reproduce the figures is available upon reasonable request.

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

## Acknowledgements

The authors would like to thank Nicholas Chin of Yale-NUS College for assistance in extracting US census data, and Barkley Dai of Yale College for compiling an early version of the SUHI United States SUHI Explorer tool in Google Earth Engine. This work was supported by a National University of Singapore Early Career Award to A.H. (Grant Number: NUS_ECRA_FY18_P15) and Samuel Centre for Social Connectedness (Grant number: AWDR14157).

## Author contributions

All authors contributed equally to the conceptualization and design of this work, analyzed data, and wrote the paper. T.C. led development of the SUHI dataset.

## Competing interests

The authors declare no competing interests.
