## [Peer Review File · Nature Communications]

Reviewer Comments, first round -

Reviewer #1 (Remarks to the Author):

This article not only presents a quality mapping of the intermixing between racial and economic inequalities related to UHI exposure but comes at a perfect timing when racial (and associated economic) inequalities come at the forefront of societal discussions in the US and abroad.

The paper is nicely presented, with compelling data and figures. The well-balanced discussion between economic and racial factors provide a great contribution to the literature on social impacts of climate change and adaptation strategies in cities (which by the way will exacerbate UHI and its incidence inequalities - I suggest the authors briefly mention and discuss this).

My major criticism is related to the non consideration of age as a critical factor for the impacts of extreme heat. In principle higher income neighborhoods or localities have higher life-longevity statistics and larger aged-population. Pregnant women and toddlers are other risk groups after aged people. In that sense I would think it is important to explicitly differentiate exposure from sensitivity and adaptive capacity; the combination of these three defines vulnerability (sensu IPCC). Then the combination of vulnerability with hazard provides a risk assessment. As such the title could read "Urban Heat Island EXPOSURE/SENSITIVITY/VULNERABILITY inequalities...", or at least the proper word should be in the abstract. That said, I do not think sensitivity and adaptive capacity are completely addressed with the employed dataset, since it considers race and income but not other variables such as age that can influence the impacts of heat extremes.

One other comment is regarding the possibility of validation of the presented results. What does official statistics of excess deaths or hospitalizations in extreme heat episodes say? Is there a racial and income prevalence?

I suggest the authors to revise the manuscript in respect and judge whether it would be possible to include other variables in their analysis that influence sensitivity to extreme heat, and whether available data corroborates the presented results.

David M. Lapola

Reviewer #2 (Remarks to the Author):

The authors used a database they developed to map urban heat island inequities in the U.S. It is troublesome that the authors do not set up hypotheses to test but apparently use the data to support a priori conclusions.

The authors do not analyze climate change so there is no reason to use limited space to devote the introductory paragraph to information not relevant to the analyses. Further, it would be better to not include speculations on vicious cycles that also are not relevant to the results. It would be better to use that space to provide more information on the methods.

While it is understandable for authors to want to frame their work as unique, it is quite a stretch to say that multiple studies of UHI and inequities in different cities do not provide pervasive evidence of the magnitude of these inequities.

"Major urban areas" should be shown in a map in SM. The authors fail to provide basic information on and justification for analytic choices. One of several examples is why compare individuals below the poverty line with those 2x above the poverty line? This creates the impression the authors are only reporting statistically significant results.

Support is needed for assumptions such as UHI intensities are larger in boreal and tropical areas.

The analyses need to be conducted by climate zones to limit confounding from other weather variables, such as humidity.

There also needs to be comparison of the UHI across summer months: are the UHI the same in May and August?

There are enough city-level publications on UHI that it should be possible to validate the results for some cities.

At some point, text implied the analyses were only for summer months, but that is not clear.

Lines 111-115 are confused. It has been known for decades that temperature-mortality relationships are J-shaped but how is that relevant to a study of summer temperatures? There is significant controversy about how cold-related mortality could change with climate change -- and that is not relevant either. This information does not support the claim that the distribution not just the mean of UHI is important -- although it is true the distribution is important. A better explanation is needed of exactly what was meant.

Lines 117-120 are obtuse. Who prefers higher mean temperatures with little dispersion vs lower temperatures with more dispersion? If these are individual preferences, then please cite literature to support this contention.

Line 121: how was "desirability" determined? As this is central to the analyses, much more explanation is required.

The discussion is very weak; it does not compare the results with results from similar studies and fails to fully explore limitations, such as using one-year of cross-sectional data. What does it mean for the results that satellites over-estimate land surface temperatures? There is no evidence to support the implicit assumption that people live, work, and play only within their census tract.

It was frustrating to attempt to look up references for further information, only to find that key references are under submission. Further, many references are missing. The authors are strongly suggested to update their literature review. Further, references is rather random in the sense that multiple citations do not support the statements where they are listed.

Summary of Major Changes:

- Included age as an additional sociodemographic variable to consider differences in surface UHI intensity exposure in major U.S. cities. Specifically, we evaluated SUHI intensity for the very young (less than 5 years) and the elderly (age 65 and above), and also by race (i.e., white, black, people of color populations less than 5 years and over age 65).
- Made our research question and hypothesis regarding the systematic differences in SUHI exposure for different sociodemographic groups in major U.S. cities explicit, and added significance thresholds for all comparisons between mean SUHI intensity exposures between groups (i.e., white vs. all people of color, etc.).
- Made clearer the conceptual framework of our analysis by referencing three components: hazard -- measures of the spatial distribution of a potential harm, for our case SUHI intensity; exposure -- the intersection of the spatial distribution of human populations with the hazard; and vulnerability -- the propensity to suffer damage when exposed to the hazard. We aligned our results along these components and divided our results into 3 separate tables to make it clear.
- Added a new figure that shows the density of SUHI intensity values for population sub-groups for cities in our analysis to more clearly show the main aim of our study - to compare differences in these groups' exposure to SUHI intensity.
- Clarified and expanded language regarding the inequality framework in the Methods section.
- Expanded on the study's limitations in the discussion section to address reviewers' comments.

Reviewer #1 (Remarks to the Author):

This article not only presents a quality mapping of the intermixing between racial and economic inequalities related to UHI exposure but comes at a perfect timing when racial (and associated economic) inequalities come at the forefront of societal discussions in the US and abroad. The paper is nicely presented, with compelling data and figures. The well-balanced discussion between economic and racial factors provide a great contribution to the literature on social impacts of climate change and adaptation strategies in cities (which by the way will exacerbate UHI and its incidence inequalities - I suggest the authors briefly mention and discuss this).

Thank you for this nice feedback. We wholeheartedly agree with you that this work contributes to the discussion of climate change's social impacts and adaptation strategies in cities. Although this is not the main focus of our study and reviewer 2 has cautioned us against making too many references to climate change directly, as our analysis only covers one cross-sectional time period, we have added some framing along the lines of vulnerability to heat stress in the introduction and application to climate mitigation (and hence adaptation) strategies in the discussion.

Lines (416-420): “Currently disadvantaged groups suffer more from greater heat exposure that can further exacerbate existing inequities in health outcomes and associated economic burdens, leaving them with fewer resources to adapt to increasing temperature (Islam et al., 2017).”

My major criticism is related to the non consideration of age as a critical factor for the impacts of extreme heat. In principle higher income neighborhoods or localities have higher life-longevity statistics and larger aged-population. Pregnant women and toddlers are other risk groups after aged people. In that sense I would think it is important to explicitly differentiate exposure from sensitivity and adaptive capacity; the combination of these three defines vulnerability (sensu IPCC). Then the combination of vulnerability with hazard provides a risk assessment. As such the title could read "Urban Heat Island EXPOSURE/SENSITIVITY/VULNERABILITY inequalities...", or at least the proper word should be in the abstract. That said, I do not think sensitivity and adaptive capacity are completely addressed with the employed dataset, since it considers race and income but not other variables such as age that can influence the impacts of heat extremes.

We responded to this helpful comment in several ways. First, due to the wide range of the way common terms are used in the literature, we recognized the need to clearly articulate a conceptual framework of risk analysis in order to let the reader know which components we address in this paper:

Lines 68-83: ““The primary focus of this study is testing whether different sociodemographic groups have different exposure to SUHI intensity, whether, for instance, people of color or low-income groups have a different exposure to SUHI intensity relative to their white or upper income peers. Conceptually, a risk analysis of heat related stress includes three components: hazard -- measures of the spatial distribution of a potential harm, for our case SUHI intensity; exposure -- the intersection of the spatial distribution of human populations with the hazard; and vulnerability -- the propensity to suffer damage when exposed to the hazard (see, for example, Cardona et al., 2012 and Lapola et al., 2019).”

As suggested, we also changed the title to emphasize our focus on the exposure, with the understanding the dataset can be useful for future studies that examine the vulnerability from the epidemiological perspective.

We thank you for the suggestion to consider age as a vulnerability factor in SUHI intensity. Although we do not have the data to conduct a comprehensive vulnerability analysis, we are able to extend our main results to understand whether the conclusions change if we consider a particularly vulnerable subset (defined by age) of the population. We replicate our analysis for sensitive age groups over 65 and under 5, as well as for these age categories by race

(non-Hispanic white, non-Hispanic black, and all people of color). We find that particularly the Under 5 population is on average exposed to higher Surface UHI intensity (2.2 ± 2.68) compared to 2.06 ± 2.72 degrees C for the average population. Over 65, however, are actually exposed to slightly lower SUHI intensity (1.84 ± 2.72). As the reviewer suggests, the results by age and race reinforce the same patterns we see when only evaluating differences in SUHI intensity by race alone. Both young and older populations of color are exposed to higher SUHI than their white peers. We added these data in Table 3 and for each city in Table S. We discuss these results at length in a new “Vulnerability” subsection, lines (236-264).

One other comment is regarding the possibility of validation of the presented results. What does official statistics of excess deaths or hospitalizations in extreme heat episodes say? Is there a racial and income prevalence?

In regards to this point, since we do not have health outcome to heat related death data at the same levels of spatial aggregation, it is harder to cross validate these results. However, we have included some heat-related mortality data to the introduction, as a means of validating our decision to consider certain sociodemographic groups as you suggest below as well as reviewer 2, although we agree with you that we do not have enough data to fully develop an index similar to your study (Lapola et al., 2018) that can assess them fully.

Specifically, lines 297-340 in the discussion, we note: “The Intergovernmental Panel on Climate Change (IPCC) has identified the “increasing frequency and intensity of extreme heat, including the urban heat island effect” as a relevant hazard for certain age groups (i.e., elderly, the very young, people with chronic health problems), which creates a risk of increased morbidity or mortality during extreme heat periods (IPCC, 2014). While relating the SUHI disparities we observe across U.S. cities to possible health outcomes is difficult, owing to both prevalence of confounding factors in these populations groups, as well as the differences between LST and more comprehensive metrics of heat stress (Anderson et al., 2013), there is evidence of disparities in heat-related health outcomes across the United States and for individual cities (Rosenthal et al., 2014; Vaidyanathan et al., 2020). For instance, Rosenthal et al. (2014) found positive correlations between heat-related mortality rates and income for neighborhoods in New York City. More recently, based on national-level data, Vaidyanathan et al., 2020 found higher heat-related mortality rates among non-Hispanic American Indians/Alaska Natives and Black Americans than for non-Hispanic Whites.

In addition to evaluating the general scope of potential heat-related environmental inequality concerns, the metrics developed in our study can identify precisely in which cities specific sociodemographic groups are most adversely exposed to surface UHI intensity and the risk of heat-related health effects for vulnerable groups. These data can thereby assist policy makers in designing interventions to mitigate this exposure differential, as well as facilitating analysis of different scenarios to select the most appropriate strategy to mitigate UHI exposure in an equitable manner.”

We also reviewed the Centers for Disease Control and Prevention (CDC) official Heat-related Deaths (2004-2018) report and added a few sentences and cited this report in the introduction to justify our selection of the sociodemographic indicators that we considered (race, income, and age) due to the higher death rates for non-Hispanic white and elderly populations, although the CDC report does not include mortality rates to heat exposure by income or poverty status.

Specifically, in the introduction we have made reference to these statistics:

Lines 92-93: “Between 2004 and 2018, 39 percent of heat-related deaths occurred in ages 65 years or older, and the highest heat-related death rates occurred in non-Hispanic white populations (Vaidyanathan et al., 2020).”

Lines 107-110: “These 175 largest U.S. cities cover approximately 65 percent of the total population and are also where most U.S heat-related deaths have occurred in the last 15 years (Vaidyanathan et al., 2020).”

I suggest the authors to revise the manuscript in respect and judge whether it would be possible to include other variables in their analysis that influence sensitivity to extreme heat, and whether available data corroborates the presented results.

We appreciate your helpful suggestion and have revised our manuscript to include age as an additional variable for analysis and have found that available mortality data in the U.S. confirms elderly populations (age 65 and older) comprise a greater proportion of heat-related deaths compared to other age groups.

Reviewer #2 (Remarks to the Author):

The authors used a database they developed to map urban heat island inequities in the U.S. It is troublesome that the authors do not set up hypotheses to test but apparently use the data to support a priori conclusions.

While the primary aim of our paper was descriptive, intended to describe disparities in surface UHI intensity among sociodemographic groups in major U.S. cities, we acknowledge that the paper is strengthened by presenting clear hypotheses in the paper’s framing and introduction. We therefore have included some text in the Introduction (Lines 68-83) to clarify the aim of our study to better set up the paper’s main hypothesis, which is that systematic differences do exist in SUHI intensity between sociodemographic groups in major U.S. cities.

“The primary focus of this study is testing whether different sociodemographic groups have different exposure to SUHI intensity, whether, for instance, people of color or low-income groups have a different exposure to SUHI intensity relative to their white or upper income peers. Conceptually, a risk analysis of heat related stress includes three components: hazard -- measures of the spatial distribution of a potential harm, for our case SUHI intensity; exposure -- the intersection of the spatial distribution of human populations with the hazard; and vulnerability

-- the propensity to suffer damage when exposed to the hazard (see, for example, Cardona et al., 2012 and Lapola et al., 2019).”

In addition to the difference in means tests for SUHI and Inequality indexes in all the tables, we have modified the maps to identify those cities in which the difference in mean exposure by demographic groups are statistically significant.

The authors do not analyze climate change so there is no reason to use limited space to devote the introductory paragraph to information not relevant to the analyses. Further, it would be better to not include speculations on vicious cycles that also are not relevant to the results. It would be better to use that space to provide more information on the methods.

We have removed references to climate change and wording on vicious cycles from the introduction. We have also provided more information about our methodological choices, including the reason for considering the sociodemographic characteristics of age, race and income, as well as the decision to focus on the 175 major U.S. cities with populations above 250,000. Based on the first reviewer’s comments, however, we have added a discussion on how heat exposure in cities can be understood in terms of climate related-hazard and age-related indicators of vulnerability. Although the IPCC has used this framework to frame the potential consequences of heat islands in future climate, it is not exclusive to climate change, being a standard framing tool in the vulnerability literature, and can be used to understand these different components of risk and their consequences on human health in the present climate.

While it is understandable for authors to want to frame their work as unique, it is quite a stretch to say that multiple studies of UHI and inequities in different cities do not provide pervasive evidence of the magnitude of these inequities.

We did not intend to make the assertion that our research is unique in seeking to understand the distributional impacts of surface UHI intensity. Rather, we reference previous studies, which are predominantly focused on single or a few cities, to extend such analyses to understand whether the inequitable distributional patterns of surface UHI intensity are pervasive and hold across a large sample size, in our case, all major U.S. cities with greater than 250,000 in population.

We have made more clear references to previous studies that examine UHI inequities in different cities (Voelkel et al., 2018; Chakraborty et al., 2019; Clinton et al., 2013; Li et al., 2017; Madrigano et al., 2015; Lin et al, 2012; Uejio et al., 2011; and Johnson et al., 2009) albeit on a smaller scale, and clarified the aim of our paper in the introduction.

We now emphasize that it is important to consider such a large sample since the cities chosen to be the subject of individual case studies may not be nationally representative.

lines(55-59): “Case studies may also reflect selection bias, i.e. prior beliefs regarding inequitable distributions of heat exposure may have motivated such scientific inquiry for particular locations, such that the chosen cities may not be representative of the nation as a whole.”

In addition, the one study that examines a relatively large set of US cities, Hoffman et al 2020, examines the disparities in heat exposure for redlined neighborhoods defined in the 1930s, not disparities in exposure for humans that incorporates the substantial demographic changes of the past 80 years.

Lines (31-42): “Hoffman et al. (2020) find that neighborhoods that were “redlined” for minorities for 108 U.S. cities in the 1930s have summer surface temperature profiles that are significantly higher than other coded residential areas. In light of the substantial demographic changes and urban growth patterns over the past 90 years, however, the extent to which this finding translates into current racial or income disparities remains unclear. “

"Major urban areas" should be shown in a map in SM. The authors fail to provide basic information on and justification for analytic choices. One of several examples is why compare individuals below the poverty line with those 2x above the poverty line? This creates the impression the authors are only reporting statistically significant results.

In Figure 2, each point represents one of the 175 major urban areas, but for additional clarity, we have added a map in the Supplementary Materials (Figure S2) that shows the location and extent of all of the major urbanized areas used in this study, also shown below:

We added more explanation in the introduction and methods sections to explain our methodological choices, including the addition of age as suggested by reviewer 1. For example, we mention in the introduction that we consider elderly populations above the age of 65, considering it is the age group with the highest percentage of heat-related mortalities in the U.S. (39 percent of total deaths). We also explain why we consider the 175 largest U.S. cities with populations over 250,000, since they cover approximately 65 percent of the total population and are also where most U.S heat-related deaths have occurred in the last 15 year (Vaidyanathan et al., 2020).

With respect to the exclusion of analysis of populations 1 to 2 times above the poverty line, the American Community Survey includes population data for individuals at several intervals below 2 times the poverty line and includes one category for 2 times above the poverty line. Rather than include each of these categories, for ease of exposition, we sought to investigate the tails of the distribution: the poor (those below poverty) and the relatively rich (above 2 times). We did not intend to selectively include or bias our results. We now include results for the missing gap (1-2 times the poverty line) in Tables 1 and 2. When we consider populations that are 1 to 2 times above the poverty line, similar patterns exist - wealthier individuals have lower mean surface UHI intensity, though the value for the intermediate group (2.5 ± 2.67 degrees C) is closer to populations below poverty than 2 times above (1.8 ± 2.72 degrees C).

Support is needed for assumptions such as UHI intensities are larger in boreal and tropical areas.

We apologize if our wording was unclear - we do not a priori assume that UHI intensities are larger in boreal and tropical areas. Instead, one of our findings (see Table 1 and Table S4) is that we observe larger UHI intensities in cities that are located in boreal and tropical areas. We have modified the beginning of that sentence to make it clear that we are not making this assumption but rather we are reporting an observation from analyzing our data.

Lines 126-135: “Panel a of Table 1 describes differences in exposure to SUHI by population groups defined by race/ethnicity and income (see Methods for demographic group definitions) for urbanized areas in each climate zone. For total population, summer day SUHI intensity is lowest ($0.40 \pm 1.75^\circ\text{C}$) in Arid zones, potentially due to the presence of more vegetation in urban areas compared to their rural references, which moderates the urban-rural LST differential or SUHI intensity (Chakraborty et al., 2019; Chakraborty et al., 2019). Most cities are in Boreal and Temperate zones, with a mean SUHI intensity of 2.2°C .”

The analyses need to be conducted by climate zones to limit confounding from other weather variables, such as humidity.

While we do present our results disaggregated according to climate zones (Tables 1-3 and Table S4), we agree that there are a range of variables, including humidity, air temperature, wind speed, etc., that are relevant to heat stress and that land surface temperature (LST) is a proxy. We have included this point in the Discussion: Limitations section to acknowledge the limitations of SUHI intensity as a proxy for urban heat exposure.

“Heat stress also depends on factors other than the surface UHI and air temperature, including humidity, wind speed, and radiation (Oleson et al., 2015). SUHI intensity, however, is still an important proxy that can predict air temperature and heat stress (Chakraborty et al., 2020).”

There also needs to be comparison of the UHI across summer months: are the UHI the same in May and August?

We apologize for our previous lack of clarity and have clarified in the Introduction and the Methods sections that by summer months, we mean June, July and August, which comprises the northern hemisphere summer. Surface UHI intensity during these months is roughly identical, as seen from the density plot below. Given this relative consistency across the summer month, we have kept our analysis focused on the period as a whole and added this figure (Figure S1) to the supplementary file.

There are enough city-level publications on UHI that it should be possible to validate the results for some cities.

We have added references in the discussion section to several studies that examine UHI-related disparities for different sociodemographic groups and are consistent with our findings (i.e., Uejio et al., 2011 found in Philadelphia and Phoenix that areas with higher black populations experienced higher heat-related mortality).

Lines 281-285: “These findings provide comprehensive evidence supporting the narrative presented by earlier case studies that minority and low income communities bear the brunt of the UHI effect (Chakraborty et al., 2019; Hoffman et al., 2019; Harlan et al., 2006; Huang et al., 2011; Johnson et al., 2009; Uejio et al., 2011) [...]”

At some point, text implied the analyses were only for summer months, but that is not clear.

We have made it explicit in the Introduction (Lines xxxx) that our analysis is focused on the summer months of June, July, and August when mean temperatures are higher than other times of the year for cities in the United States and when the incidence of heat waves is also higher.

“We narrow our analysis to the summer months of June, July and August when the UHI intensity is most pronounced during the day and when mean temperatures are generally higher than other periods through the year (Johnson et al., 2009).”

Lines 111-115 are confused. It has been known for decades that temperature-mortality relationships are J-shaped but how is that relevant to a study of summer temperatures? There is significant controversy about how cold-related mortality could change with climate change -- and that is not relevant either.

We apologize for the imprecise wording. By non-linear we meant that the adverse outcome-temperature relationship was increasing and convex above moderate temperatures -- not that there is a decreasing relationship for cold temperatures and increasing relationship for warm temperatures. We have clarified the relevant text in the introduction to mean that incremental increases in SUHI in areas that are already hot may be at a higher risk of heat-related impacts than those with lower average temperatures, which would make the impacts at higher temperatures more damaging:

This information does not support the claim that the distribution not just the mean of UHI is important -- although it is true the distribution is important. A better explanation is needed of exactly what was meant.

We have clarified these lines in the introduction to mean that incremental increases in SUHI in areas that are already hot may be at a higher risk of heat-related impacts than those with lower average temperatures, which would make the impacts at higher temperatures more damaging:

“Recognizing that health impacts of summer heat exposure are likely to be non-linear (Seppanen et al., 2006; Deschenes et al., 2011; Graff et al., 2016; Wang et al., 2017), i.e., incremental increases in environmental heat load may lead to disproportionately higher risk (Johnson et al., 2009), we also consider environmental inequality metrics that evaluate the importance of within-group inequalities with respect to SUHI spatial distribution and exposure for different sociodemographic groups.”

Lines 117-120 are obtuse. Who prefers higher mean temperatures with little dispersion vs lower temperatures with more dispersion? If these are individual preferences, then please cite literature to support this contention.

In mathematical terms we have the following. Suppose that in the summer health damage is an increasing convex function of temperature (a 1 degree temperature increase causes more damage at higher temperatures) then, all else equal, Jensen's inequality implies that the average health damage for a population in which everyone faces an identical temperature exposure will be lower than that of a population that has the same mean exposure but an unequal temperature distribution. Therefore, for any unequal temperature distribution there exists a more desirable (from a health perspective) distribution characterized by a higher mean and no inequality.

We rewrote the relevant text to try to convey this point without using mathematical jargon. We cut the material in the introduction to the sentence appearing in the response to the previous comment. We introduce more detail in the “Intra-group variation in exposure”. Here we briefly describe the intuition behind how the non-linear damage function implies a mean-dispersion trade off:

Lines 204-215: “A potential drawback to focusing on average exposures by demographic group is that they can mask the existence potential ‘hotspots’, geographic areas in which individuals are exposed to elevated levels of the hazard. Hotspots are particularly problematic when comparing exposures across groups if a 1 degree temperature increase causes more damage at higher temperatures. In such cases, even if two groups were to hypothetically face the same average exposure, a group in which half individuals were exposed to a temperature of, say, 38C and half were exposed to 32C, would suffer higher adverse effects than a group in which all individuals were exposed to 35C.”

We then introduce the KP index as a measure of inequality, and describe the results of this part of the analysis.

In the Methods:Inequality Metrics section, we provide more detail about the ethical framework underlying the KP index and reference its use in the literature evaluating distributions of income and environmental harm.

Lines 551-584: “The goal of comparing exposure levels across population groups is to determine whether a distribution of UHI intensities for a given group is preferable in some sense to that of another. In contrast to approaches identifying correlations between summer temperatures and neighborhood characteristics such as historical redlining (Hoffman et al., 2019) or percentage poor or low income (e.g., Voelkel et al., 2018), we place the unit of analysis on the individual to better understand human welfare implications of surface UHI exposure.

There is no clear link between what individuals find desirable and the significance of statistical correlations between neighborhood attributes. It is theoretically possible, for example, for the average individual in a demographic group to be better off with a positive (versus negative) correlation between summer heat and their group's majority status in a neighborhood if most members of the group happen to live in neighborhoods in which they are a minority.

A simple individual-based metric such as mean exposure is potentially misleading, however, due to non-linear adverse health impacts of summer heat -- evidence suggests that above a moderate threshold the incremental damage of a one-degree increase grows as the temperature gets hotter (Seppanen et al., 2006; Deschenes et al, 2011; Graff and Shrader 2016; and Wang et al, 2017). That is, a perfectly equal summer temperature distribution is generally preferable to an unequal distribution with the same mean.

We adapt an ethical framework commonly used to study income distributions to compare distributions of environmental harm (Sheriff and Maguire, 2020) Under this framework a distribution is considered more desirable than another if it would be chosen by an impartial agent who knows only that she will receive an outcome from that distribution but is ignorant regarding what that outcome will be. [...].”

Line 121: how was "desirability" determined? As this is central to the analyses, much more explanation is required.

See above response and elaborated section on Inequality metrics in the Methods section.

The discussion is very weak; it does not compare the results with results from similar studies and fails to fully explore limitations, such as using one-year of cross-sectional data. What does it mean for the results that satellites over-estimate land surface temperatures? There is no evidence to support the implicit assumption that people live, work, and play only within their census tract.

We have improved the discussion section to elaborate our findings in relationship to previous studies, as well as to address reviewer 1's comment about heat-stress due to the urban heat island effect as a metric of vulnerability to climate change. We also moved the section on the study's Limitations, which was previously at the end of the Methods section, to the Discussion section to make the study's shortcomings more apparent and prominent in the discussion.. Specifically, we mention the limitations regarding the use of census tracts as the main spatial unit of analysis and comparison, which assume that people live in their census tracts and therefore are exposed to surface UHI intensity levels within those tracts. We lack data, however, on where people work and spend leisure time, which are likely in different areas and could expose individuals to much different levels of SUHI than in the census tracts where they reside.

We also address the limits in using satellite-derived measures of LST for SUHI calculations, which are generally higher than air temperature measurements particularly during the daytime (Lines 451-460):

“Our analysis relies on satellite-based estimates, which could overestimate UHI magnitude compared to in situ weather stations, particularly for daytime measurements (Zhang et al., 2014), when shade from tree canopies or buildings reduce air temperature in a way that is not captured from a satellite's vantage point. Our estimates, therefore, likely slightly overestimate the absolute measures of UHI (in °C), but in lieu of dense, widely available ground-based air temperature networks, satellite-derived estimates represent the best data source.”

Further, at this scale, we do expect that the census data we use is representative of present conditions, which is also an implicit assumption in most studies that use census data. The 5-year UHI intensity is calculated corresponding to and generally covering that same time interval.

It was frustrating to attempt to look up references for further information, only to find that key references are under submission. Further, many references are missing. The authors are strongly suggested to update their literature review. Further, references is rather random in the sense that multiple citations do not support the statements where they are listed.

Thank you for pointing out these issues in the references. For the references that are under submission, one has been accepted (Chakraborty et al., 2020), which we have replaced with the updated citation. Others we have corrected and we have re-reviewed all of the citations to ensure that those we cite support the claims that are made in the appropriate sentences.

Reviewer Comments, second round -

Reviewer #1 (Remarks to the Author):

I applaud the authors for the effort to include the age analysis in their study and considerations about validation of the developed index. The manuscript is ready for publication from my point of view.

Reviewer #2 (Remarks to the Author):

Thank you for the extensive responses to the reviewer comments. However, it would have been helpful for readers if more of the explanations were included in the publication and not just in the responses.

I remain concerned about the results that you label as being for boreal areas. How did you define boreal areas? Boreal in North America is defined as extending south to 55N. Detroit is at 42.3N. Youngstown is 41.1. According to <<https://en.wikipedia.org/wiki/K%C3%B6ppen_climate_classification>> these cities and others are not in a boreal climate. I did not check whether other cities are actually within the claimed climate zones, but clearly they need to be.

The discussion section is improved but still includes new results and still repeats the Results section. The discussion also contains multiple unsupported normative statements. One example is that older adults may choose to live in greener areas. The following sentence states there also is evidence and then goes on to another topic. No evidence is provided. Referring to urban and rural areas does not provide support.

The fact that globally consistent data were used does not mean the data are useful for decision-making at the local level. Co-production studies typically conclude that such datasets are not necessarily relevant.

REVIEWER COMMENTS

Reviewer #1 (Remarks to the Author):

I applaud the authors for the effort to include the age analysis in their study and considerations about validation of the developed index. The manuscript is ready for publication from my point of view.

Thank you.

Reviewer #2 (Remarks to the Author):

Thank you for the extensive responses to the reviewer comments. However, it would have been helpful for readers if more of the explanations were included in the publication and not just in the responses.

We have gone back through our responses to reviewer comments from the first round of review to ensure that each point we addressed in the response is reflected in the manuscript. Where specific line numbers or sections were not mentioned in our initial response, we went back and ensured that explanations were adequately reflected in the manuscript itself.

For instance, with respect to the reviewer's comment with respect to why we highlighted those 2x above the poverty line, we responded, that for the ease of exposition, we sought to investigate the tails of the distribution, the poor (those below poverty) and the relatively rich (above 2 times), but we went back and added results for the additional ACS category of individuals 1-2 times above the poverty line in Tables 1 and 2. We have added this information in the Methods (Lines 557-558) to make this point clear.

Lines 557-558: "While results for each of these income categories is provided in our Tables, for the ease of exposition, we focus our discussion on the tails of the income distribution: the poor (those below poverty) and the relatively rich (above 2 times)."

The only other response comment that was not originally reflected in the manuscript was in response to your comment regarding temperature preferences. We also now include a brief discussion of Jensen's inequality in the methods section when introducing the inequality index approach:

Lines 583-597: "A simple individual-based metric such as mean exposure is potentially misleading due to non-linear adverse health impacts of summer heat. Evidence suggests that above a moderate threshold damage is an increasing convex function of temperature, i.e., a 1 degree temperature increase causes more damage at higher temperatures (50-53). In such cases, Jensen's inequality implies that, all else equal, the average health damage for a population in which everyone faces an identical summer heat exposure will be lower than that of a population with the same mean exposure but an unequal temperature distribution. It follows

that for any unequal temperature distribution there exists a more desirable (from a health perspective) distribution characterized by a higher mean and no inequality. That is, a perfectly equal summer temperature distribution is generally preferable to an unequal distribution with the same mean.

I remain concerned about the results that you label as being for boreal areas. How did you define boreal areas? Boreal in North America is defined as extending south to 55N. Detroit is at 42.3N. Youngstown is 41.1. According to https://en.wikipedia.org/wiki/K%C3%B6ppen_climate_classification these cities and others are not in a boreal climate. I did not check whether other cities are actually within the claimed climate zones, but clearly they need to be.

We apologize for this confusion. The link the reviewer has provided is for the Köppen climate classification, while we have used the Köppen-Geiger climate classification, which identifies the following climate zones for the U.S.: equatorial, arid, warm temperate, and snow (see: <http://koeppen-geiger.vu-wien.ac.at/usa.htm>). We had originally switched the 'snow' climate zone with the term 'boreal' based on a more recent paper by the same authors of the classification scheme (http://koeppen-geiger.vu-wien.ac.at/pdf/Paper_2017.pdf): 'The boreal climate is a synonym for the historically introduced term snow climate and, in a global context, the alpine climates are called polar climates.'

To avoid this confusion, we have reverted back to the original naming scheme for the climate zones as defined in the Köppen-Geiger classification. All references to the 'boreal' climate zone have been replaced with 'snow,' 'tropical' has been replaced with 'equatorial.' In tables and figures we abbreviate 'warm temperate' to 'temperate' due to space limitations and note this slight abbreviation in the text:

Lines 128-130: "We group urbanized areas by Köppen-Geiger (54) climate zones: arid, snow, warm temperate (henceforth referred to as temperate), and equatorial"

The discussion section is improved but still includes new results and still repeats the Results section. The discussion also contains multiple unsupported normative statements. One example is that older adults may choose to live in greener areas. The following sentence states there also is evidence and then goes on to another topic. No evidence is provided. Referring to urban and rural areas does not provide support.

We were unsure exactly which new results the reviewer was referring to, but we moved the illustrative examples of Greenville and Baltimore counties to the Results section so that we are not referencing new figures in the Discussion section. Specifically, we added 'Section D: Illustrative Examples' to the Results section and moved the paragraph discussing these two cities and references to Figure 3 there.

With regards to the writing of the Discussion section, we were not sure exactly which statements were considered repetitive, so we took care to ensure that any repetitive statements were removed. Because the main purpose of this section was to elaborate and contextualize our Results, we elected to keep some references to our main findings to set up the discussion points.

For example, in the first paragraph of the Discussion section (starting Line 285), we kept one sentence succinctly summarizing our Results as the core main finding of our analysis: “We find the distributions of summer daytime SUHI intensity, taking into account both the mean and dispersion, is worse for both people of color and the poor, compared to white and wealthier populations in nearly all major U.S. cities. As we illustrated in Figure 2, this pattern holds not only at the national level, but in almost all major urban areas regardless of geographical location or climate zones, with a particularly intense difference in the Northeast and upper Midwest of the continental U.S.”

And in lines 302-307, the first several sentences of the second paragraph, we edited to avoid repeating the Results too much but to set up discussion for why we found less average SUHI intensity exposure for elder populations than what we expected, given the heat-related mortality rates for this population subgroup: “Although age presents a vulnerability to SUHI and elderly individuals aged 65 and older comprise a substantial percentage (39 percent) of heat-related deaths in the U.S. (Vaidyanathaen et al., 2020), our finding that populations over 65 are on average slightly less exposed (1.84 C versus 2.06C for those under 65) could have several explanations.”

We then edited the following sentences in that same paragraph regarding elder populations to avoid the sentence that may have appeared normative (i.e., previously “older populations may choose to live in areas of cities with more greenery” has been removed). We support our observation that older populations are not more exposed to higher SUHI intensity by referencing a Harvard study that found that a substantial segment of populations over 65 live in suburban, less dense areas where research has found, and by definition, are typically greener than built-up and denser urban environments, with the exception of arid climates where rural areas are often desert areas. (Lines 307-315): “Because SUHI intensity and greenness (as measured by Normalized Difference Vegetation Index or NDVI) are negatively correlated (Chakraborty et al., 2020), cooler areas tend to be greener. There is evidence that older populations over the age of 65 tend to live in suburban areas in the U.S. Approximately half live in rural areas or in urban areas with less than 1 housing unit per acre, and 28 percent live in suburban areas (Joint Center for Housing Studies of Harvard University, 2016), which are typically greener than denser urban areas, except in arid climates (Chakraborty et al, 2019b; Nitoslawski et al., 2016; and Hansen et al., 2005).”

We also removed a sentence after Line 393 that repeated a finding in the Results regarding our finding that only in 1 city (McNally, TX) white populations have a higher exposure than those living in poverty. (Sentence from previous version now removed from Discussion: “By

comparison, in only one city do white populations have a higher average exposure than those living below poverty (Table S1).)

The fact that globally consistent data were used does not mean the data are useful for decision-making at the local level. Co-production studies typically conclude that such datasets are not necessarily relevant.

Thank you for this comment, we agree that there are specific factors at the local level (e.g., urban form, building materials, stakeholder preferences, planning and design processes) that are relevant in the consideration of SUHI mitigation and management and that globally consistent datasets are unable to measure or take this into consideration. Similar critiques have been raised regarding the utility of satellite-derived globally consistent SUHI data for decision-making and planning for 'climate-sensitive design' at the city scale. See Martilli, A., Roth, M., Chow, W. T., Demuzere, M., Lipson, M., Krayenhoff, E. S., ... Hart, M. A. (2020, June 20). Summer average urban-rural surface temperature differences do not indicate the need for urban heat reduction. <https://doi.org/10.31219/osf.io/8gnbf> and the original paper we cite Manoli et al., 2019 and Manoli's response: Manoli, G., Fatichi, S., Schläpfer, M., Yu, K., Crowther, T. W., Meili, N., ... Zeid, E. B. (2020, June 24). Reply to Martilli et al. (2020): Summer average urban-rural surface temperature differences do not indicate the need for urban heat reduction. <https://doi.org/10.31219/osf.io/mwpna>.

Therefore, from Lines 362-376, we have removed the statement regarding our dataset and analysis's utility for SUHI mitigation "tailored for local conditions" and modified it as such, including a referencing to Manoli et al. (2019) who have also caveated the limits of a globally consistent dataset for UHI management in cities around the world.

Lines 365-376: "Decision-makers and urban planners can utilize this information as a starting point to identify best practices and strategies for mitigating the SUHI and the inequalities in its distribution, although there are certainly localized, contextually-specific factors that must be considered when determining SUHI management strategies. As Manoli et al. (2019), who used similar globally-consistent data to evaluate drivers of SUHI in 30,000 cities around the world, acknowledge, these data can provide a 'first-order' analysis to understand base-level SUHI exposures and differences to complement more fine-grained data on local factors that influence the SUHI (see Limitations section for more discussion on data issues).

Co-production is indeed a relevant approach used in urban planning centered in the active involvement and engagement of actors in the production of knowledge in topics such as urban forestry, urban development, waste management and climate adaptation (Campbell et. al 2016; Frantzeskaki, N., & Kabisch, N. 2016; Gutberlet 2015; Satorras et.al 2020). Furthermore, recent literature exemplifies that this type of data could be useful as a starting point or as inputs in co-production initiatives, alongside or supporting locally-generated datasets. (Iwaniec et.at 2020, Anenberg et.al 2020). However, the scope of this research is not to address the relevance of this dataset in this type of approach, hence it is not specifically mentioned in the discussion.

Reviewer Comments, third round -

Reviewer #2 (Remarks to the Author):

Thank you for addressing most of my comments.

For your information, "normative" does not mean repetitive. A normative statement expresses a value judgment about whether a situation is desirable or undesirable. Whereas a descriptive statement is meant to describe the world as it is, a normative statement is meant to talk about the world as it should be.

This is what I referred to when mentioning that several sentences in the Discussion were normative without references.

Also, although your dataset can not address co-production, it is still important to discuss that effective interventions are co-produced; e.g. in order to tailor the results for local levels, co-production is needed.

We would like to thank the editor and referees for the time and effort put into helping improve the manuscript. Below we provide detailed responses to the last round of comments. Since our response refers to previous rounds of comments, for clarity we use black font for the referee's current comments, red font for the referee's previous comments and blue font for our response. Our previous responses we highlight in a green font.

Reviewer #2 (Remarks to the Author):

Thank you for addressing most of my comments.

For your information, "normative" does not mean repetitive. A normative statement expresses a value judgment about whether a situation is desirable or undesirable. Whereas a descriptive statement is meant to describe the world as it is, a normative statement is meant to talk about the world as it should be. This is what I referred to when mentioning that several sentences in the Discussion were normative without references.

There were two relevant aspects to the reviewer's last round of comments that we consider here:

1. Discussion section repeats the Results section.
2. Unsupported normative statements

In the last set of reviews, regarding repetition the referee stated:

"The discussion section is improved but still includes new results and still repeats the Results section."

To which we responded:

"With regards to the writing of the Discussion section, we were not sure exactly which statements were considered repetitive, so we took care to ensure that any repetitive statements were removed. Because the main purpose of this section was to elaborate and contextualize our Results, we elected to keep some references to our main findings to set up the discussion points.

For example, in the first paragraph of the Discussion section (starting Line 285), we kept one sentence succinctly summarizing our Results as the core main finding of our analysis: "We find the distributions of summer daytime SUHI intensity, taking into account both the mean and dispersion, is worse for both people of color and the poor, compared to white and wealthier populations in nearly all major U.S. cities. As we illustrated in Figure 2, this pattern holds not only at the national level, but in almost all major urban areas regardless of geographical location or climate zones, with a particularly intense difference in the Northeast and upper Midwest of the continental U.S.

...

And in lines 302-307, the first several sentences of the second paragraph, we edited to avoid repeating the Results too much but to set up discussion for why we found less average SUHI intensity exposure for elder populations than what we expected, given the heat-related mortality rates for this population subgroup: “Although age presents a vulnerability to SUHI and elderly individuals aged 65 and older comprise a substantial percentage (39 percent) of heat-related deaths in the U.S. (Vaidyanathaen et al., 2020), our finding that populations over 65 are on average slightly less exposed (1.84 C versus 2.06C for those under 65) could have several explanations.”

Regarding “normative” statements, the referee commented:

“The discussion also contains multiple unsupported normative statements. One example is that older adults may choose to live in greener areas. The following sentence states there also is evidence and then goes on to another topic. No evidence is provided. Referring to urban and rural areas does not provide support.”

To which we responded:

We then edited the following sentences in that same paragraph regarding elder populations to avoid the sentence that may have appeared normative (i.e., previously “older populations may choose to live in areas of cities with more greenery” has been removed). We support our observation that older populations are not more exposed to higher SUHI intensity by referencing a Harvard study that found that a substantial segment of populations over 65 live in suburban, less dense areas where research has found, and by definition, are typically greener than built-up and denser urban environments, with the exception of arid climates where rural areas are often desert areas. (Lines 307-315): “Because SUHI intensity and greenness (as measured by Normalized Difference Vegetation Index or NDVI) are negatively correlated (Chakraborty et al., 2020), cooler areas tend to be greener. There is evidence that older populations over the age of 65 tend to live in suburban areas in the U.S. Approximately half live in rural areas or in urban areas with less than 1 housing unit per acre, and 28 percent live in suburban areas (Joint Center for Housing Studies of Harvard University, 2016), which are typically greener than denser urban areas, except in arid climates (Chakraborty et al, 2019b; Nitoslowski et al., 2016; and Hansen et al., 2005).”

The referee did not indicate, and we could not find, other instances of “normative” statements requiring additional support.

Also, although your dataset can not address co-production, it is still important to discuss that effective interventions are co-produced; e.g. in order to tailor the results for local levels, co-production is needed.

Thank you for this comment - we agree and have added the following sentence:

Line (370-373): “Studies have demonstrated the importance of co-production (i.e., involving citizens in the production of knowledge and planning decisions) in developing tailored urban environmental policies (Satorras et al., 2020).